# Acute and Chronic Pain from Facial Skin and Oral Mucosa: Unique Neurobiology and Challenging Treatment

**DOI:** 10.3390/ijms22115810

**Published:** 2021-05-28

**Authors:** Man-Kyo Chung, Sheng Wang, Se-Lim Oh, Yu Shin Kim

**Affiliations:** 1Center to Advance Chronic Pain Research, Department of Neural and Pain Sciences, School of Dentistry, Program in Neuroscience, University of Maryland at Baltimore, Baltimore, MD 21201, USA; swang1@umaryland.edu; 2Department of Advanced Oral Sciences & Therapeutics, School of Dentistry, University of Maryland at Baltimore, Baltimore, MD 21201, USA; soh@umaryland.edu; 3Health Science Center at San Antonio, Programs in Integrated Biomedical Sciences, Department of Oral & Maxillofacial Surgery, School of Dentistry, Translational Sciences, Biomedical Engineering and Radiological Sciences, University of Texas, San Antonio, TX 78229, USA

**Keywords:** chronic pain, mucosa pain, orofacial pain

## Abstract

The oral cavity is a portal into the digestive system, which exhibits unique sensory properties. Like facial skin, the oral mucosa needs to be exquisitely sensitive and selective, in order to detect harmful toxins versus edible food. Chemosensation and somatosensation by multiple receptors, including transient receptor potential channels, are well-developed to meet these needs. In contrast to facial skin, however, the oral mucosa rarely exhibits itch responses. Like the gut, the oral cavity performs mechanical and chemical digestion. Therefore, the oral mucosa needs to be insensitive, to some degree, in order to endure noxious irritation. Persistent pain from the oral mucosa is often due to ulcers, involving both tissue injury and infection. Trigeminal nerve injury and trigeminal neuralgia produce intractable pain in the orofacial skin and the oral mucosa, through mechanisms distinct from those seen in the spinal area, which is particularly difficult to predict or treat. The diagnosis and treatment of idiopathic chronic pain, such as atypical odontalgia (idiopathic painful trigeminal neuropathy or post-traumatic trigeminal neuropathy) and burning mouth syndrome, remain especially challenging. The central integration of gustatory inputs might modulate chronic oral and facial pain. A lack of pain in chronic inflammation inside the oral cavity, such as chronic periodontitis, involves the specialized functioning of oral bacteria. A more detailed understanding of the unique neurobiology of pain from the orofacial skin and the oral mucosa should help us develop novel methods for better treating persistent orofacial pain.

## 1. Introduction

Orofacial skin and the oral mucosa protect the body from physical and chemical damage, infection, dehydration, and heat loss. Even though both oral mucosa and facial skin are covered by highly specialized stratified epithelia, the two tissues are structurally different in many ways: hair follicles and sweat glands exist in the skin, while the oral mucosa surrounds the teeth and contains taste buds and minor salivary glands. The oral mucosa is more permeable than skin. Nonkeratinized mucosa, such as the floor of the mouth and the buccal mucosa, is more permeable than other regions of the oral mucosa, and transmucosal drug delivery is under active development [1]. As the oral mucosa heals after injury faster than skin and without scar tissue, the cellular, molecular, and immunologic differences between oral mucosa and skin have been widely studied, and the oral mucosa has been used as a model for developing methods for scarless cutaneous healing [2]. Therefore, distinct sensations arising from the oral mucosa and facial skin have drawn much attention [3,4].

As a portal into the digestive system, the oral cavity is exposed to a dynamic environment featuring mechanical, thermal, and chemical stimuli due to the ingestion and mastication of various foods. The oral mucosa exhibits sensory properties, similar to both facial skin and the gut. Like facial skin, the oral mucosa requires an exquisite level of sensitivity to mechanical, thermal, and chemical stimuli, in order to detect the properties of foods and to prevent the ingestion of harmful materials. Pain from the oral mucosa also modulates jaw movements and masticatory activities [5]. At the same time, similar to the gut, the oral mucosa needs to be somewhat insensitive to stimuli, in order to resist the mechanical mastication of hard food or to endure the swallowing of hot drinks. When ingested food is perceived to be unpleasant, the food is spit out of the oral cavity for protection. In addition to toxic food materials, multiple etiologies cause acute or chronic pathological pain in the oral cavity. Oral pain critically affects the quality of life, as it degrades vital functions, such as eating and swallowing, especially when the pain is chronic [6]. Here, we review the characteristics of the chemosensory and somatosensory functions of the oral mucosa, as well as its neurobiological mechanisms, in comparison to those in facial skin. We also review the pathological conditions inducing acute or chronic oral and facial pain and discuss their underlying mechanisms. A better understanding of the neurobiological mechanisms of oral and facial pain should help in the development of more effective methods for managing the associated conditions, eventually improving the quality of care for patients. Although orofacial pain is derived from different tissues due to a diverse etiology, including an autonomic function (Table 1), in this study, we focused on several subtypes of pain from the oral cavity and face that are more difficult to diagnose or treat.

## 2. Physiological Somatosensation and Pain from Oral Mucosa and Facial Skin

Peripheral nociception in orofacial tissues occurs at the peripheral branches of the trigeminal nerves—the fifth cranial nerves (Figure 1). Trigeminal ganglia harbor neuronal cell bodies of sensory neurons, projecting to both the peripheral and central sides. Nociceptive nerves synapse with second-order neurons within the trigeminal nucleus complex, especially the caudal area [7], which relays the nociceptive signals to the brain regions involved in sensory discriminative and affective pain.

### 2.1. Somatosensation of Oral Mucosa and Facial Skin

The thermal and mechanical sensitivity of the oral mucosa differ from that of facial skin. Different intraoral sites also show different sensitivities. Extraoral skin and the tongue tip are generally more sensitive than the gingival mucosa, as has been shown by quantitative sensory testing (QST) [9] (Figure 2). The face and tongue are more sensitive to cold, warmth, and mechanical stimuli than the gingiva. The heat pain threshold is also higher in the gingiva, while the mechanical pain and vibration detection thresholds are lower in the tongue than other tissues. The pressure pain threshold is lowest in the tongue and highest in the face. There is no difference among the oral mucosa and facial skin in terms of the cold pain threshold. Interestingly, paradoxical heat sensations upon cooling do not occur in the skin or tongue but occurred in the gingiva of 71% of subjects, suggesting a poor thermal discrimination ability of the gingiva [9]. Other reports have also shown similar tendencies, with minor differences. Tactile sensitivities of the lower lip, anterior tongue, and buccal mucosa are greater than that of the soft palate, posterior tongue, and posterior pharyngeal wall [10]. The tip of the tongue is the most sensitive, in terms of tactile sensitivity, followed by the hard palate, lateral tongue, buccal mucosa, and gingiva [11]. The pressure pain threshold is greater in the maxillary gingiva than the mandible but showed no differences at sites along the tooth rows [12]. Cold sensitivity at the tongue tip is higher when compared to the chin, whereas warmth and heat pain sensitivity are lower [13]. Responses to cold and warm stimuli are poorer in the oral mucosa than in the supraorbital skin and nasal mucosa [14].

The oral mucosa is well-developed to detect various chemical stimuli with considerable sensitivity. The topical application of capsaicin to the oral mucosa produces a burning sensation, where capsaicin sensitivity is similar between the supraorbital skin and the oral mucosa but less than in the nasal mucosa [14]. The topical application of capsaicin on the oral mucosa led to strong pain from the subsequent heat stimulus both at the site of application (primary hyperalgesia) but also outside the site of application (secondary hyperalgesia) [15]. In addition to capsaicin, mustard oil and cinnamaldehyde also enhance the pain caused by the application of heat (49 °C) to the tongue [16]. In contrast, subjects who frequently eat foods containing capsaicin show a slightly higher warmth detection threshold than subjects who rarely eat foods containing capsaicin, suggesting desensitization by the chronic ingestion of capsaicin [17]. Mustard oil and cinnamaldehyde also modestly enhance the pain caused by the application of cold to the tongue. Menthol ingestion increases both the warm and cold detection thresholds [17] but does not affect the pain intensity in response to subsequent heat or cold stimuli [15].

### 2.2. Properties and Projections of Primary Afferents in Oral Mucosa and Facial Skin

Microneurographic studies in humans have shown that oral mucosal mechanoreceptors exhibit both fast and slow adapting properties [18]. The stimulation of fast-adapting fibers produces sensations of vibration, whereas the stimulation of slow-adapting fibers produces sensations of constant pressure. Unlike facial skin, however, there are no fast-adapting mechanoreceptors with the properties of Pacinian corpuscle afferents [18]. In the oral mucosa of experimental animals, three types of mechanoreceptors exist: Meissner corpuscles, Ruffini endings, and Merkel cells. Free nerve endings serve as thermoreceptors or nociceptors [19]. In mice, Merkel cells, Meissner corpuscles, and glomerular corpuscles have been found in the tongue, hard palate, and gingival mucosa, but no Pacinian corpuscles or Ruffini endings were found [20]. The density of Merkel cells in the hard palate decreases in aged mice [20]. The functional properties of oral mucosal nociceptive afferents are not well-known. In Wistar rats, using a single unit recording of a lingual nerve innervated to the mandibular gingival mucosa [21], it has been found that 46% of fibers are non-nociceptive low-threshold mechanoreceptors, and 54% of fibers are classified as nociceptive. Based on their conduction velocity in the oral mucosa, the nociceptive fibers can be classified as Aδ (58%) or C fibers (42%). Based on their responses to thermal, mechanical, and chemical stimuli, oral mucosal nociceptors are classified into four types: Aδ high-threshold mechanonociceptors, Aδ mechano–heat nociceptors, Aδ polymodal nociceptors, and C polymodal nociceptors. The mechanical threshold of each of these receptors is higher than that in skin, whereas the heat threshold is similar. Unlike skin afferents, in the oral mucosa, the size of the receptive field is greater in C polymodal nociceptors than in Aδ nociceptors. Aδ polymodal nociceptors are rare in the skin but enriched in visceral afferents [21]. A recent study has also shown the characteristics of afferents innervated to the tongue in mice [22]. Approximately 50% of lingual afferents are c fibers, while approximately 30% are Aδ fibers. In this study, the implantation of oral squamous carcinoma into the tongue increased spontaneous firing, decreased the mechanical thresholds of the c and Aδ mechanoreceptors, and reduced the proportion of mechanically insensitive fibers [22].

In rodents, approximately 40–45% of buccal mucosal afferents express transient receptor potential cation channel subfamily V member 1 (TRPV1) and transient receptor potential cation channel subfamily A member 1 (TRPA1) [23]), which may transduce the burning pain caused by capsaicin and mustard oil. These afferents include peptidergic afferents, and, in the buccal mucosa, capsaicin and mustard oil evoke the release of the calcitonin gene-related peptide (CGRP) [24]. The retrograde labeling of gingival mucosa afferents in the trigeminal ganglia (TG) of rats and mice has shown that gingival mucosal afferents are small- to medium-sized, and their average size is smaller than tooth pulp afferents [25,26]. In rats, approximately 50% of gingival afferents are CGRP-positive, while 25% of gingival afferents are substance P-positive [27]; in contrast, in mice, only 23% of gingival afferents are CGRP-positive [26]. In mouse gingiva, CGRP-positive afferents are highly colocalized with TRPV1 [26]. In rats, 76% of gingival afferents are tropomyosin receptor kinase A-positive neurons, and 50% are isolectin B4-binding neurons [21]. Transient receptor potential cation channel subfamily M member 8 (TRPM8)-expressing fibers in the oral mucosa [28] likely mediate the sensation due to menthol. Menthol itself does not evoke the release of CGRP from the buccal mucosa but enhances the cold-evoked release of CGRP, which depends on TRPM8 [24]. The overexpression of neurturin—a member of the glial cell line-derived neurotrophic factor (GDNF)—in keratinocytes increases the expression of TRPM8 in TG and increases the oral sensation generated by menthol [29]. A recent study showed that the tongue mucosa and muscle are innervated by neurons expressing CGRP (25% of tongue afferents), TRPV1 (17%), 5HT3A (21%), TrkC (31%), parvalbumin (14%), NPY2R (12%), and Mrgprd (7%), which suggests the projection of peptidergic and non-peptidergic nociceptors, proprioceptors, and low-threshold mechanoreceptors [30]; however, the functional implications of these differences have not been studied well.

The ascending pathways of orofacial pain have been thoroughly reviewed elsewhere [31]. Trigeminal subnucleus caudalis (Vc) neurons are critical hubs for transmitting oral and facial pain [7], which are activated by noxious or non-noxious stimuli applied to the oral cavity, as well as the face. The effects of the lingual application of a variety of chemical stimuli, such as capsaicin, ethanol, histamine, mustard oil, nicotine, acid, and piperine, have been assessed [32]. Other regions, such as cranial nerves (C1/C2), subnuclei interpolaris (Vi), Vi/Vc, and subnuclei oralis (Vo), are also known to be involved in orofacial pain. Mapping phosphorylated extracellular signal-regulated kinase (pERK) following the injection of capsaicin into various orofacial areas showed the somatotopic arrangement of neurons in Vc and the upper cervical spinal cord and revealed distinct chemical transmission pathways between the intraoral and extraoral facial sites [33]. Capsaicin injection into the extraoral ophthalmic, maxillary, or mandibular regions induces pERK+ neurons in the ventral, middle, and dorsal portions of Vc, respectively. In contrast, the localization of pERK+ neurons by capsaicin injection into the intraoral mucosa does not show distinct segregation. Capsaicin injection into the tongue or lower gum induces pERK+ neurons localized to the dorsal half of the Vc, whereas capsaicin injection into the anterior hard palate, upper gum, or buccal mucosa induces pERK+ neurons in both dorsal and ventral Vc. Unlike extraoral injection, capsaicin injection into intraoral sites produces a large number of pERK+ neurons in contralateral Vc [33]. Oral and facial regions also show a rostrocaudal somatotopic pattern: oral regions are represented in rostral Vc, whereas the lateral face regions are represented more in caudal Vc [7].

### 2.3. Itch Sensation of Oral Mucosa Is Weaker Than That of Facial Skin

The itch sensation in the oral cavity has not been well-studied. Although an itching sensation associated with allergic reactions occurs in the oral mucosa, histamine-induced flares are limited to the skin, as histamine is not associated with the itch sensation in human oral mucosa [34]. Indeed, the injection of histamine into the mucosa at the dorsal surface of the tongue rarely produces itch-like responses in mice, whereas the same dose of histamine evokes robust scratching upon injection into the cheek skin (Figure 3). Histamine-induced itches largely depend on TRPV1 and TRPV1-expressing afferents [35]. As a capsaicin injection into the tongue mucosa produces robust nocifensive behaviors, a lack of itch response is not due to the lack of TRPV1-expressing afferent projection into the tongue mucosa. It is possible that oral mucosa afferents are devoid of itch-specific factors, such as phospholipase β3 [35]. Indeed, the tongue mucosa is rarely projected by afferents expressing MrgprA3 or the gastrin-releasing peptide, two well-established markers for itch-sensing afferents [30,36]. It is clear that further studies on the neurobiology of pruriceptors in the oral mucosa are needed.

## 3. Pathological Painful Conditions and Underlying Neurobiology

### 3.1. Ulcers and Injury of Oral Mucosa

While numerous pathological conditions can cause acute and chronic pain in the oral mucosa, one of the most common causes of pain from the oral mucosa is ulcerated lesions due to multiple etiologic factors, such as infection, autoimmune conditions, trauma, or neoplasia. Viral infections, such as Herpes simplex virus type 1, can cause painful ulcers in the oral mucosa. Some patients with COVID-19 infection have also presented painful oral lesions, such as blister or ulcers [37]. These conditions are mainly acute pain conditions but may contribute to the transition from acute to chronic. Recurrent aphthous stomatitis is the most common ulcerated mucosal lesion. Mechanical irritation by dentures is also a common cause. Radiation therapy for treating head and neck cancer or chemotherapy causes damage to the oral mucosa. Oral mucositis, an “inflammation of oral mucosa resulting from cancer therapy typically manifesting as atrophy, swelling, erythema and ulceration” [38], typically involves an extensive area of the oral mucosa over a prolonged period of time (>70 days). Oral mucositis is one of the most debilitating adverse effects of cancer therapy, which significantly lowers the quality-of-life of patients [38]. The current management of oral mucositis and pain depends on the opioid or palliative care, which is not satisfactory [39]. An improved understanding of the pathophysiology of oral mucositis should help to develop more specific and efficacious treatments.

The role of reactive oxygen species and proinflammatory cytokines, such as tumor necrosis factor (TNF) and interleukin 1β, in conjunction with the innate immunity in the pathogenesis of mucositis have been extensively studied [40]. In rats, oral mucositis has been modeled by the application of acetic acid alone or in combination with 5-fluorouracil, a chemotherapeutic agent, on the labial mucosa [41,42]. In ulcerated mucosa, the mucosal barrier is breached, and bacterial loading is increased, along with the infiltration of immune cells. Spontaneous pain and mechanical hyperalgesia in ulcerated oral mucosa involve prostaglandins and protease-activated receptor 2 signaling, leading to the sensitization of TRPV1, TRPA1, and TRPV4 [41,42,43]. Radiation of the oral cavity produces glossitis in mice, accompanied by decreased burrowing activity and increased face-wiping behaviors [44]. In irradiated mice, dissociated TG neurons showed increased activation by histamine and capsaicin, suggesting a role of TRPV1 [44]. Incisions in the buccal mucosa increase head withdrawal in response to heat, cold, and mechanical stimuli [23]. Such hypersensitivity is accompanied by the increased expression of TRPV1 and TRPA1 in mucosal afferents. The pharmacological inhibition of TRPV1 reverses the heat hypersensitivity, whereas the inhibition of TRPA1 reverses the cold and mechanical hypersensitivity [23].

### 3.2. Oral Cancer

Oral squamous cell carcinoma (OSCC) is the most frequent malignancy in the oral mucosa, where pain is one of the frequent symptoms. Although pain and discomfort are apparently obvious in advanced lesions with ulcers and masses, pain is often the first symptom in patients with OSCC [45,46]. Indeed, oral cancer patients report greater spontaneous pain and functional restriction from pain, compared to subjects with normal mucosa or precancerous lesions, suggesting that pain is an important predictor of the cancer transition [47]. Patients with recurrent OSCC show even greater pain [48].

OSCC implantation to the gingiva in rodents produces mechanical and thermal hyperalgesia, suggesting the sufficiency of OSCC for producing pain [49]. The mechanisms by which cancer cells lead to pain have been widely studied, and the interactions of cancer cells with immune cells and nociceptive afferents produce pain through mechanisms distinct from the canonical inflammatory or neuropathic pain conditions [50]. Multiple factors, such as endothelin, nerve growth factor (NGF), adenosine triphosphate, proton, and proteases, are released from cancer cells and attract immune cells to eventually induce the peripheral sensitization of nociceptive afferents [50]. The inoculation of SCC cells into the tongue leads to peripheral sensitization [22] and central sensitization in Vc [51]. More studies are warranted, in order to obtain a detailed understanding of the unique peripheral and central molecular mechanisms of oral cancer pain.

### 3.3. Neuropathic Pain

#### 3.3.1. Trigeminal Neuralgia and Painful Post-Traumatic Trigeminal Neuropathy

Trigeminal neuralgia [8] is characterized by recurrent sudden electrical shock-like attacks in the distribution of the trigeminal nerve. TN is rare in the U.S. population: 5.9 per 100,000 women and 3.4 per 100,000 men. The pain often involves both the facial skin and the intraoral mucosa, where paroxysmal oral pain can be triggered by innocuous stimuli, such as tooth brushing, mouth opening, talking, or chewing. In some patients, sour or spicy solutions trigger paroxysmal pain [52]. TN occurs spontaneously in combination with vascular compression of the trigeminal nerve root entry zone (classical TN) or secondary to other diseases, such as multiple sclerosis (secondary TN). Painful post-traumatic trigeminal neuropathy (PTTN) occurs following craniofacial or oral trauma. PTTN is apparently more prevalent than trigeminal neuralgia: following a trigeminal nerve injury, 3–5% of patients develop chronic pain [53]. An injury of the lingual nerve or inferior alveolar nerve during the third molar extraction or injection of local anesthetic is the most common iatrogenic cause of PTTN [54]. The pain is moderate to severe, exhibiting a burning and shooting quality, and is usually continuous, lasting most of the day. Unlike TN, PTTN rarely shows the triggering of pain [53]. Most patients with PTTN present sensory dysfunction, such as hypesthesia, paresthesia (abnormal nonpainful sensation), dysesthesia (unpleasant sensation), or allodynia, whereas TN patients rarely show sensory dysfunction in QST. The differences between TN and PTTN are summarized in Table 2. An anticonvulsant, such as carbamazepine, is recommended as the first line of treatment for TN, whereas a tricyclic antidepressant, such as amitriptyline, is used for PTTN. PTTN patients are more resistant to therapy than patients with TN [55] or other neuropathic pain, such as postherpetic neuralgia or diabetic neuropathy [56]. Interestingly, in patients with intraoral pain, a local anesthetic spray shows immediate relief of pain in 67% of TN patients [57], suggesting the substantial contribution of peripheral mechanisms. Botulinum toxin type A has also shown analgesic efficacy in TN, following an extraoral or intraoral injection [58,59]. The topical administration of capsaicin to the oral mucosa improves the intraoral chronic neuropathic pain [60]. In a patient with PTTN in craniofacial regions, persistent pain could be attenuated by topical capsaicin [61]. As topical capsaicin produces analgesia primarily through peripheral mechanisms [62], these studies suggest a potential critical contribution of peripheral nociceptors to PTTN and TN. This notion has been supported by a series of preclinical studies, as summarized below (Figure 4). This peripheral treatment can be used as an alternative treatment for chronic neuropathic pain from the facial skin and oral mucosa, when the first-line treatment using an anticonvulsant or antidepressant is not effective or cannot be used due to adverse side effects.

The neurobiological mechanisms underlying trigeminal neuropathic pain have been determined using multiple preclinical rodent models. Most animal models of trigeminal neuropathic pain involve direct injury to the peripheral branches of trigeminal nerves, such as the infraorbital nerve [63,64], and mimic the characteristics of PTTN. A rat model with a compressed trigeminal nerve root mimics the painful conditions from TN: focal demyelination of the trigeminal nerve root, prolonged mechanical and cold allodynia, and the suppression of pain by carbamazepine [65]. Recent thorough reviews of the myriad central and peripheral neurobiological mechanisms in rodent models are available [66,67]. Here, we highlighted several differential mechanisms of trigeminal neuropathic pain, compared to spinal area pain, particularly the indispensable role of peripheral nociceptors in the maintenance of chronic neuropathic pain (Figure 4). Injuries to trigeminal nerves cause different gene regulations within TG compared with the effects of injury to the spinal nerve in the dorsal root ganglia (DRG) [68], suggesting potentially different contributions of the primary afferents to chronic pain. Following injury, the genes encoding cannabinoid receptor 2, metabotropic glutamate receptor 5, 5-hydroxytryptamine receptor 1a, and tachykinin receptor 1 are upregulated in TG at 5 days, while the genes encoding adenosine receptor 1, catechol-O-methyltransferase, and TNF are upregulated in TG at 21 days; all of these genes are downregulated in DRG. Interestingly, CGRP receptor antagonists reduce the mechanical allodynia in rats with chronic constriction injury of the infraorbital nerve but not of the sciatic nerve [69], reflecting the differential contributions of neuropeptides to the maintenance of neuropathic pain in facial versus spinal areas. The role of primary afferents in the maintenance of trigeminal neuropathic pain has been further strengthened through studies of the long-lasting analgesia of localized capsaicin injection into the facial skin [63]. Nerve injury to the maxillary V2 nerves enhances the functions of primary afferent neurons not only in the V2 but also the V3 area (Figure 5A–D). As the intraganglionic spreading of the sensitizing signal occurs within TG, extraterritorial secondary hyperalgesia in the craniofacial area can occur through peripheral mechanisms at the level of TG [64]. Mechanical hyperalgesia from a trigeminal nerve injury is also maintained by descending facilitation through serotonergic projection from rostroventromedial medulla, which sensitizes the central terminals of TRPV1-expressing afferents within Vc [70] (Figure 5E–J). Large-diameter trigeminal afferents in the TG or central terminals terminated in the deep layers in Vc become more sensitive to capsaicin after injury, suggesting the increased expression of TRPV1 in presumable non-nociceptive neurons, which are very important functional changes, as neuralgic attacks in patients with TN are usually triggered by a tactile stimulation that is normally non-noxious [70]. Compared to hind-paw pain, facial pain induces a greater activation of the parabrachial nuclei, which is associated with affective pain [71]. Surprisingly, a subpopulation of TG neurons is monosynaptically connected with the parabrachial nucleus; this connection likely mediates a stronger affected pain from the orofacial area [71]. In addition to the primary afferents, the potential contribution of sympathetic nerves also differs. In contrast to a spinal nerve injury, which leads to sprouting of the sympathetic nerves in DRG, a nerve injury in the trigeminal nerves leads to sympathetic nerve sprouting only at the affected skin but not in TG [72,73].

#### 3.3.2. Atypical Odontalgia

Atypical odontalgia (AO) (post-traumatic trigeminal neuropathy, persistent idiopathic dentoalveolar pain, or phantom tooth pain) is a chronic pain condition located in the teeth after endodontic treatment or in the dentoalveolar site after tooth extraction, with no signs of pathology on clinical or radiographic examination [75]. The diagnosis and treatment of AO is challenging, where a misdiagnosis can lead to unnecessary endodontic treatment or extraction of teeth. AO has been proposed to involve the peripheral neuropathy. In intraoral QST, approximately 90% of AO patients show sensory abnormalities: the most frequent symptoms were an increased painful mechanical and cold sensation and increased cold and mechanical detection [75]. This hyperalgesia may be attributable to the central mechanisms, suggested by the fact that capsaicin administration to the gingiva produces greater pain in AO patients [76]. Capsaicin application, either ipsilateral or contralateral to AO-associated pain, shows increased pain, but the threshold to electrical stimulation of the peripheral nerves was not different from healthy subjects [76]. Furthermore, the injection of lidocaine to the site of pain only partially relieves spontaneous pain, suggesting a contribution of the central mechanisms of sensitization [77]. AO is also suggested to be a chronic overlapping pain condition: AO occurs more frequently in females and is often comorbid with other chronic pain conditions, such as temporomandibular disorders (TMD; more frequent in females) or headaches, and emotional disorders, such as depression and anxiety [78].

### 3.4. Burning Mouth Syndrome

Burning mouth syndrome (BMS) is defined as an “intraoral burning or dysesthetic sensation, recurring daily for more than 2 h per day over more than 3 months, without evident causative lesions on clinical examination and investigation” (The Orofacial Pain Classification Committee 2020). BMS occurs overwhelmingly more frequently in women, especially in postmenopausal women. BMS frequently occurs with comorbid chronic pain conditions, such as headaches, TMD, back pain, and fibromyalgia [79]. BMS patients exhibit more psychological symptoms—including somatic symptoms, anxiety and insomnia, social dysfunction, and severe depression—than healthy subjects [80].

Although the etiology of BMS is not clear, the nerve fiber density in the tongue mucosa is lower than that in healthy controls, suggesting the involvement of small fiber neuropathy in BMS [81,82,83]. The decreased density of nerve fibers appears to correlate with the extent and duration of the pain [82,84]. In BMS patients, the TRPV1 expression is increased in the remaining nerve fibers and the NGF expression is increased in the nerve fibers and basal epithelial layer. The self-reported pain level in BMS patients correlates with the density of the TRPV1-expressing fibers [83]. This may explain the hypersensitivity to the lingual application of capsaicin in BMS patients [85]. Furthermore, nerve fibers expressing P2 × 3, an ionotropic purinergic receptor enriched in nociceptors, are also increased in the tongue mucosa of BMS patients [81]. These results suggest that peripheral neuropathy is associated with BMS pain. However, the sensory phenotypes of BMS patients do not solidly support this notion. In QST, the thermal and mechanical sensations and pain of intraoral mucosa or extraoral skin in BMS patients are highly variable among studies, and it is difficult to conclude that BMS patients show abnormal somatosensation, compared to healthy subjects [79]. A meta-analysis, however, showed a lower cold detection threshold (i.e., increased cold sensitivity) and a higher warmth detection threshold (i.e., reduced warmth sensitivity) in BMS patients, compared to healthy controls [86]. The source of such variations is unknown. It is possible that only a subpopulation of BMS patients is associated with peripheral neuropathy. Even among the BMS patients likely associated with neuropathy, symptoms of peripheral neuropathy can manifest differently in different patients. A more detailed characterization of a larger cohort of patients is necessary for a better diagnosis and treatment. For example, a topical treatment using clonazepam or capsaicin does not work in all BMS patients [8,87], suggesting differential contributions of the primary afferents to the maintenance of chronic pain in different patients. A sensitivity to the lingual application of capsaicin is reduced in patients with a longer duration of BMS pain [88], suggesting that nociceptive afferents have a reduced contribution to pain in this population, or, perhaps, there is degeneration of nociceptive afferents in BMS. It has been suggested that BMS should be classified into three categories, based on pathophysiology: peripheral small fiber neuropathy, trigeminal lesions in the periphery or brainstem, and hypofunctional dopaminergic system [89,90]. Patients with a dominant central etiology do not respond to local treatments and are often associated with psychiatric comorbidities (depression or anxiety), whereas patients with a peripheral etiology should respond better to peripheral lidocaine and topical clonazepam [90].

BMS is an idiopathic pain condition without a clear etiology; yet, to our knowledge, animal models representing entire aspects of BMS are not available. Rather, researchers have developed models for testing specific hypotheses of BMS pain. In BMS patients, artemin, a member of the glial cell line-derived neurotrophic factor, is increased in the tongue mucosa [91]. In mice, a treatment with trinitrobenzene sulfonic acid induces noninflammatory heat hyperalgesia in the tongue, accompanied by an increased expression of TRPV1 in TG, as well as an increased expression of GDNF, NGF, TNF, and interleukin 1β in the tongue mucosa. An increased expression of artemin in the oral mucosa in a transgenic mouse line in which artemin is overexpressed under the keratin promoter is sufficient to upregulate TRPV1 and TRPA1 in TG and to increase the responses to capsaicin and mustard oil [92]. The neutralization of artemin in the tongue mucosa reduces both the heat hyperalgesia and TRPV1 upregulation [91]. More preclinical studies are warranted in order to investigate the neurobiology of BMS.

## 4. Unique Contributors to Regulation of Oral Pain

### 4.1. Role of Gustatory Nerves in Pain Modulation

The oral mucosa is innervated not only by trigeminal afferents conveying somatosensation but also by gustatory nerves, the chorda tympani nerve, and the glossopharyngeal nerve (Figure 1). Taste is known to modulate pain; for example, intraoral sucrose is effective for reducing procedural pain in infants [93]. The analgesic effects of sucrose on pain have also been reported in adults. Sweet foods or sucrose increases the tolerance to pressure pain or cold pain [94]. Intraoral sucrose increases the pain threshold in a cold pressor test and reduces the activation of the brain regions, such as the anterior cingulate cortex, insula, and thalamus, associated with cold pain [95,96,97]. Intraoral sucrose reduces the pain unpleasantness induced by cutaneous phasic electric stimulation [98]. Conversely, chronic pain conditions affect taste sensations. Patients with TMD pain report decreased taste compared to healthy subjects [99]. In BMS patients, hypofunction of the chorda tympani nerve and reduced taste sensation were reported [88,100]. Chorda tympani nerve dysfunction affects not only taste but also somatosensory functions [101]. Sucrose reduces the burning pain induced by the lingual application of capsaicin in humans, where subjects with chorda tympani nerve resection show greater capsaicin perception [102]. These reports suggest supraspinal interactions of the gustatory and nociceptive circuits and that dysfunctional gustatory afferents can modulate oral pain.

The analgesic effects of oral sucrose have been investigated in rodents. In neonatal rats, sucrose transiently decreases the thermal pain in uninflamed paws and robustly reduces thermal hyperalgesia in an inflamed paw [103]. This effect only occurs in the forepaw (not in the hind paw) and is accompanied by a decreased expression of Fos—an immediate early gene and surrogate marker of neuronal activity—in the cervical, but not lumbar, spinal cord. Oral sucrose analgesia does not require neural circuits confined to the forebrain but does involve descending pain modulation; intraoral sucrose elicits Fos expression in the nucleus tract solitarius, the main central projection of gustatory nerves, and also in the periaqueductal gray and the nucleus raphe magnus [104]. The long-term intake of saccharin for 2 weeks produces analgesia for heat pain in adult Wistar rats [105]. Such sweet-induced analgesia is reduced by the systemic inhibition of cholinergic receptors, the μ opioid receptor, the 5HT2A serotonergic receptor, or the α1 adrenergic receptor [106,107]. Analgesia for thermal pain and inflammatory thermal hyperalgesia by oral treatment with sucrose in adult rats has also been demonstrated in Sprague–Dawley rats [108]. In this study, μ opioid receptor, dopamine receptor, or α2 adrenergic receptor inhibition had no effect on analgesia, but a cannabinoid 1 receptor antagonist reduced analgesia [108]. Importantly, the study showed that analgesia occurs only when the rats develop a conscious preference for sucrose (i.e., hedonic drinking) rather than a simple taste experience, suggesting the involvement of supraspinal mechanisms. Indeed, the sucrose intake does not affect the heat-evoked Fos expression in the spinal cord [108]. This is consistent with the lack of effects of gustatory stimuli on Vc neuronal activation by the noxious stimulation of tongue [109]. Although the effects of gustatory nerve damage on pain have not been studied in detail, the bilateral resection of chorda tympani nerves in rats does not affect the responses to the lingual application of capsaicin for up to 12 months [110]. The antinociceptive and potential pronociceptive effects of the gustatory nerves in oral mucosal pain need to be determined further. Another complexity of the oral mucosa is that natural compounds, such as capsaicin and menthol, can activate not only trigeminal nerves but also gustatory nerves. In humans, capsaicin and menthol induce a bitter taste [74]. Indeed, approximately 10–20% of geniculate ganglia neurons express TRPV1 and TRPA1, and a few neurons express TRPM8 [111,112]. Injury of the chorda tympani nerves upregulates the expression of TRPV1 and TRPA1 in geniculate ganglia [112]. These results suggest that the responses to intraoral capsaicin in patients with altered taste (e.g., BMS) need to be interpreted as neuroplasticity, having occurred in both the gustatory and somatosensory pathways in the peripheral and central nervous systems. The therapeutic effects of capsaicin for pain in BMS can also be attributable to the desensitization of TRPV1-expressing gustatory nerves.

### 4.2. Chronic Periodontitis without Persistent Pain

Chronic periodontitis is an infectious chronic inflammatory condition, resulting in the destruction of the periodontium. Unlike other chronic inflammatory conditions, such as osteoarthritis, which are often accompanied by persistent pain, patients with chronic periodontitis rarely experience pain until the disease reaches severe advanced stages. Among the patients with chronic periodontitis seeking treatment, only 6% reported painful gingiva [113]. A lack of pain in patients with periodontitis discourages patients from seeking timely dental treatment, eventually resulting in tooth loss. However, the neurobiological mechanisms underlying the painless progression of chronic periodontitis are not well-understood. *Porphyromonas gingivalis* is a Gram-negative bacterium, which has long been considered a keystone pathogen of periodontitis [114]. *P. gingivalis* causes the subversion of neutrophils, which leads to disruption of the host protective mechanisms and microbial dysbiosis [114]. This unique function of *P. gingivalis* in the pathogenesis of periodontitis might be related to the lack of pain. In mice, the induction of inflammation by injecting complete Freund’s adjuvant produces mechanical hyperalgesia in the gingiva. However, the inoculation of *P. gingivalis* does not produce mechanical hyperalgesia in the gingiva, and the interaction of *P. gingivalis* with macrophages prevents the sensitization of periodontal nociceptors [115]. *P. gingivalis* inoculation into the gingiva increases the macrophage expression of CXC chemokine receptor type 4 (CXCR4). The administration of a neutralizing antibody against CXCR4 leads to the development of mechanical hyperalgesia following *P. gingivalis* inoculation, which can be inhibited by a nitric oxide synthase inhibitor. The fimbriae of *P. gingivalis* bind to CXCR4 in combination with Toll-like receptor 2 in the macrophages, leading to inhibition of the antimicrobial responses [116]. CXCR4-mediated interactions of *P. gingivalis* and macrophages inhibit nitric oxide production through the activation of nuclear factor-κB [115]. Interestingly, lipopolysaccharides (LPS) from *P. gingivalis* produce analgesic effects. The injection of LPS from *P. gingivalis* into the hind paw reduces the mechanical hyperalgesia induced by hind paw incision [117]. This effect was accompanied by an increase in the anti-inflammatory cytokine IL-10. Considering that the intraplantar injection of lipopolysaccharides from *E. coli* produce hyperalgesia [118], it appears that *P. gingivalis* LPS differs from *E. coli* LPS, with respect to the capacity to modulate the pain. This is an interesting area that will need to be addressed going forward.

## 5. Conclusions

The oral cavity is a portal into the digestive system, where the oral mucosa is critical for sensitively detecting harmful food. At the same time, the oral mucosa is resistant to mechanical and thermal irritation during the mastication and ingestion of food. The mechanisms underlying the persistent pain originating in the oral mucosa have been studied in rodent models of ulcers and cancers. Studies on the mechanisms of chronic pain following trigeminal nerve injury have revealed unique mechanisms underlying trigeminal neuropathic pain, compared to the pain of spinal origins, an area that needs to be further investigated. The oral cavity is a unique environment, and modulation of the pain arising from the oral mucosa can be complicated. The mechanisms of the effects of gustatory inputs regarding the modulation of chronic pain from the oral mucosa are not well-known. Specialized functions of the oral bacteria, such as *P. gingivalis*, may lead to a lack of pain during the progression of periodontitis. A more detailed understanding of the unique neurobiology of pain from the oral mucosa should help us and is expected to aid in developing novel methods for the better treatment of mucosal pain without affecting the physiological roles of sensation and the oral functions.

## Figures and Tables

**Figure 1 ijms-22-05810-f001:**
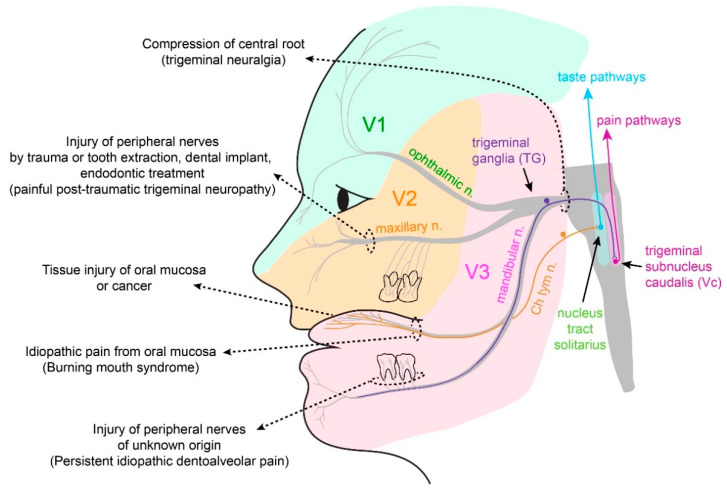
Innervation of the craniofacial sensory nerves, and the sources of persistent orofacial pain from tissue or nerve injuries. The trigeminal nerve is the fifth cranial nerve. The trigeminal ganglia (TG) contain neuronal cell bodies of sensory neurons projecting to the orofacial structures. The first branch (V1) projects to the forehead skin and dura through the ophthalmic nerve. The second branch (V2) projects to the skin, mucosa, and teeth of the upper jaw through the maxillary nerve. The third branch (V3) projects to the skin, mucosa, and teeth of the lower jaw, including the tongue mucosa. The central branches of the trigeminal sensory neurons project to the trigeminal nucleus complex in the brainstem and make synaptic connections with second-order neurons. The pain-sensing nociceptors from the orofacial area are highly connected with the caudal region of the trigeminal nucleus complex (trigeminal subnucleus caudalis; Vc). The Vc neurons relay signals to various ascending pain pathways within the brain. The tongue mucosa is also innervated by gustatory nerves, such as the chorda tympani nerve (Ch tym n), which is a part of the facial nerve (the seventh cranial nerve). Taste signals are transmitted through the chorda tympani nerve, relayed in the nucleus tract solitarius in the brainstem, and transmitted to the central taste pathways. Persistent pain from the orofacial area can be derived from multiple etiologies of injuries to the tissue or nerves. Irradiation or chemotherapy can cause oral mucositis. Oral cancer often causes pain from the oral mucosa. The direct injury of peripheral nerves due to facial trauma or tooth extraction can lead to the development of painful post-traumatic trigeminal neuropathy (PTTN). Trigeminal neuralgia [8] is a distinct entity of chronic pain, derived from compression of the central root of the trigeminal nerve. Some idiopathic chronic orofacial pain, such as burning mouth syndrome (BMS) or persistent idiopathic dentoalveolar pain, are regarded to be of neuropathic origin.

**Figure 2 ijms-22-05810-f002:**
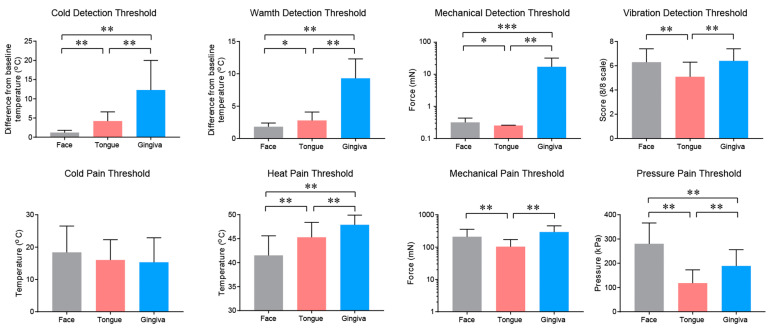
Different sensations of intraoral mucosa and extraoral skin. Results of quantitative sensory testing (QST) in 21 healthy subjects (13 women and 8 men; mean 40.4 years) on the cheek, tip of the tongue, and gingival mucosa of the upper premolar region. Mean ± SD; * *p* < 0.05, ** *p* < 0.01, and *** *p* < 0.001; paired *t*-test following Bonferroni correction for multiple comparisons. Plots were redrawn using previously published data from Pigg et al. [9], with kind permission.

**Figure 3 ijms-22-05810-f003:**
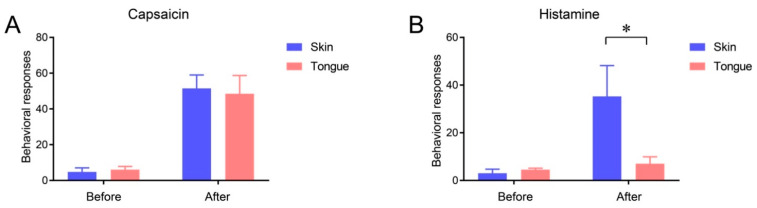
Lack of histamine-induced itch response from the tongue mucosa. The number of behavioral responses assessed 5 min before and after the injection of capsaicin (10 µg/10 µl (**A)**) or histamine (20 µg/10 µl (**B**)) into the facial skin or tongue of C57BL/6 mice. Under isoflurane anesthesia, capsaicin or histamine was injected into the facial skin subcutaneously or into the submucosa of the dorsum of the tongue. Before and after the injection of capsaicin into the facial skin, the number of wipings of the injected skin using the ipsilateral hind paw was counted. Before and after histamine injection into the facial skin, the number of scratchings of the injected site using the ipsilateral forepaw was counted. Upon the injection of capsaicin or histamine into the tongue, the number of instances of wiping, scratching, and grooming of the face using the bilateral forepaws was counted. *n* = 4 in each group. * *p* < 0.05 in post-test following two-way repeated measures ANOVA.

**Figure 4 ijms-22-05810-f004:**
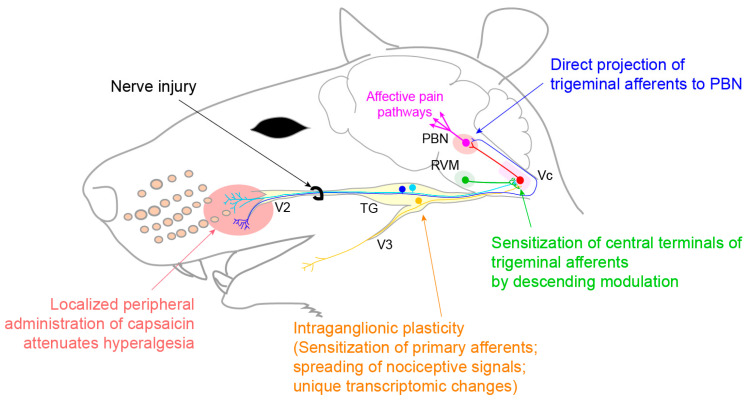
Mechanistic contribution of the nociceptive primary afferents to chronic orofacial neuropathic pain. Chronic constriction nerve injury (CCI) of the infraorbital nerve (ION), a part of V2, induces widespread unique transcriptomic changes in TG. The spreading of a nociceptive signal between the V2 and V3 neurons within TG may contribute to extraterritorial hyperalgesia. Central terminals of trigeminal afferents within the trigeminal subnucleus caudalis (Vc) are sensitized by descending facilitatory inputs from the rostral ventromedial medulla (RVM). A subset of trigeminal afferents directly projects to the parabrachial nucleus (PBN), which is a hub of affective pain pathways, without relaying at the Vc. The localized administration of capsaicin—which selectively defunctionalizes nociceptive afferents—to orofacial tissues attenuates long-lasting orofacial neuropathic pain, both in humans and rodents.

**Figure 5 ijms-22-05810-f005:**
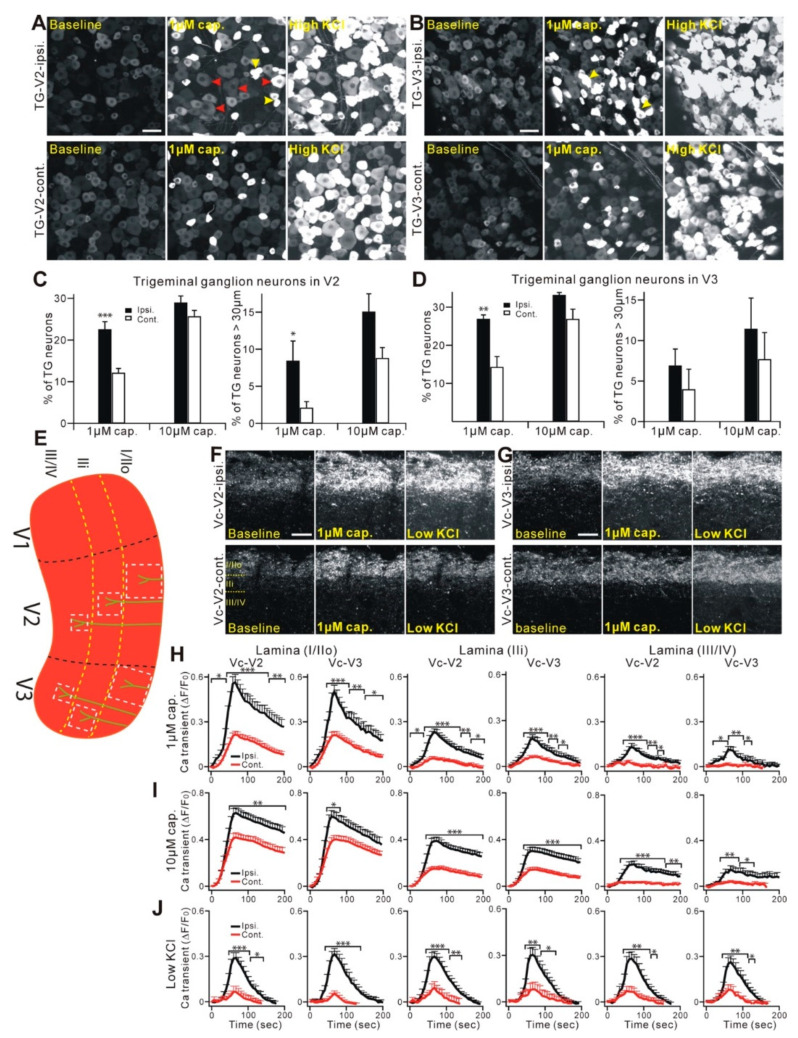
Functional plasticity changes of the trigeminal nociceptors following a craniofacial neuropathic injury. (**A**,**B**) Representative GCaMP3 imaging of TG explants from Pirt-GCaMP3 mice. TG neurons of V2 (**A**) or V3 (**B**) division after CCI-ION were activated by capsaicin (1 µM) or KCl (100 mM). Upper panels show ipsilateral a V2 (**A**) or V3 (**B**) division of TG. Lower panels show a contralateral V2 (**A**) or V3 (**B**) division of TG. Scale bar, 50 µm (for both). Arrowheads in red indicate TG neurons that are over 30 µm in diameter and activated by capsaicin. Arrowheads in yellow indicate coupled TG activation by capsaicin; that is, adjacent neurons activated together—simultaneously or in a very short time window (<1 sec). (**C**,**D**) Population data for V2 shown in (**C**) or for V3 shown in (**D**), expressed as the percentage of KCl-sensitive TG neurons. ipsi., ipsilateral; cont., contralateral. * *p* < 0.05; ** *p* < 0.01; and *** *p* < 0.001. Data are presented as the mean ± SEM. (**E**) Schematic diagram of a mouse Vc. Black dashed lines indicate the border of three large divisions (V1, V2, and V3) in Vc. White square dash lines indicate where the region of the nerve fibers and terminals [74] was selected and analyzed, in order to measure the Ca^2+^ transient at different lamina (yellow dash lines). (**F**,**G**) Representative GCaMP3 imaging of Vc slices from Pirt-GCaMP3 mice. Central fibers and terminals in the V2 (**F**) or V3 (**G**) division of Vc after CCI-ION were activated by capsaicin (1 µM) or 20-mM KCl. The upper panels show the ipsilateral V2 (**F**) or V3 (**G**) division of Vc. The lower panels show the contralateral V2 (**F**) or V3 (**G**) division of Vc. Scale bar, 50 µm. (**H**–**J**) Time course of the amplitude of the Ca^2+^ transient evoked by capsaicin (1 µM and 10 µM) or KCl (20 mM) applications at different lamina (lamina I/IIo, IIi, and III/IV) of V2 and V3 of the Vc. Capsaicin was applied in a bath from 0 to 60 s during the time course. The Ca^2+^ transient (∆F/F_0_) was normalized to the value imaged at the baseline. Ipsi., ipsilateral; cont., contralateral. * *p* < 0.05, ** *p* < 0.01, and *** *p* < 0.001. Data are presented as the mean ± SEM. The data and images were redrawn using previously published data from Kim et al. [70].

**Table 1 ijms-22-05810-t001:** Classification overview of the International Classification of Orofacial Pain (ICOP) *.

Table	Subtype
1. Orofacial pain attributed to disorders of dentoalveolar and anatomically related structures	1.1 Dental pain **1.2 Oral mucosal**, salivary gland, and jawbone pains
2. Myofascial orofacial pain	2.1 Primary myofascial orofacial pain 2.2 Secondary myofascial orofacial pain
3. Temporomandibular joint (TMJ) pain	3.1 Primary temporomandibular joint pain 3.2 Secondary temporomandibular joint pain
4. Orofacial pain attributed to lesion or disease of the cranial nerves	**4.****1****Pain attributed to lesion or disease****of****the****trigeminal****nerve** 4.2 Pain attributed to lesion or disease of the glossopharyngeal nerve
5. Orofacial pains resembling presentations of primary headaches	5.1 Orofacial migraine 5.2 Tension-type orofacial pain 5.3 Trigeminal autonomic orofacial pain 5.4 Neurovascular orofacial pain
6. Idiopathic orofacial pain	**6.1 Burning mouth syndrome (BMS)** 6.2 Persistent idiopathic facial pain (PIFP) **6.3 Persistent idiopathic dentoalveolar pain** 6.4 Constant unilateral facial pain with additional attacks (CUFPA)

* Cephalalgia, 40:129–221 (2020). **Bold**, subtype of pain focused on in this review.

**Table 2 ijms-22-05810-t002:** Comparisons of the trigeminal neuralgia [8] and painful post-traumatic trigeminal neuropathy (PTTN).

	TN	PTTN
Average Age of Onset	59	49
Etiology	Classical TN: Spontaneous, neurovascular compression Secondary TN: underlying disease (e.g., multiple sclerosis)	Trauma to facial skeleton Iatrogenic: 3rd molar extraction, injection of local anesthetic, dentoalveolar surgery, implant, endodontic treatment, orthognathic surgery
Pain	Side: Unilateral	Unilateral or bilateral (10%)
	Area involved: V3 (50%), V2 (30%), V2 + V3 (20%)	Injury-related area
	Quality: Electric shock-like pain (sometimes combined with shooting and stabbing pain)	Burning, stabbing, pressure, and throbbing
	Duration: Occurs continuously. Paroxysms lasting from a second to 2 min followed by a refractory period. Abrupt in onset and termination. Sometimes superimposed on background pain between attacks	Heterogeneous with respect to frequency and duration and often occurs continuously throughout day (50%), short attack 1–4 min (25%; longer than TN), constant, or intermediate.
	Intensity: Severe (VAS 9.1; greater than PTTN)	Moderate to severe (VAS 7.7)
	Trigger: Often triggered by innocuous stimuli within the affected region	Trigger is rarely identifiable
QST	No sensory dysfunction	Hypoalgesia, allodynia, hyperalgesia, dysesthesia, paresthesia
First line of treatment	Carbamazepine	Tricyclic antidepressant (Amitriptyline) alone or in combination with gabapentin or an SNRI (duloxetine)
Response to therapy	Pain relief greater than 50% in 74% of patients	Pain relief greater than 50% in 11% of patients

V2, maxillary branch of trigeminal nerve; V3, mandibular branch of trigeminal nerve; QST, quantitative sensory testing; VAS, visual analogue scale; SNRI, serotonin noradrenaline reuptake inhibitor. Table is created by adapting the data and description from Benoliel et al. and Haviv et al. [53,55].

## Data Availability

Not applicable.

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
