# Peer review of "Acute and Chronic Pain from Facial Skin and Oral Mucosa: Unique Neurobiology and Challenging Treatment"

_ijms, 2021, doi:10.3390/ijms22115810_

Round 1

Reviewer 1 Report

Summary

This review by Chung and colleagues gives a detailed insight into chronic pain conditions within the oralfacial area and describes preclinical and molecular findings regarding specifically chosen pain conditions. Characteristics of chemosensory and somatosensory functions of the oral muscosa and its neurobiological mechanisms in comparison to those in facial skin are discussed. Overall, this is a well summarized review outlining previous findings and ongoing challenges. The review would benefit from some more information about the current clinical standard.

Broad comments

In my view, the review would benefit from some more information about the current treatment of the introduced conditions, how successful/unsuccessful these treatments are, what kind of side effects they entail, and what kind of treatments might be benefitial in the future and/or are currently tested in clinical trials. Since pathways and mechanims are detailed in this review, the respective mechanism of action of these treaments might be touched upon as well. This might especially be benefitial for chapter 3, either as an additional paragraph to each introduced condition or as an overview table for all introduced conditions. The authors lightly touch upon that topic for the TN and PTTN. But also in light of the title of the review "...and challenging treatment", this topic should be described more thoroughly and also for all the introduced conditions. 

The abstract should be reviewed again thoroughly, optimally by a native speaker. The main text seems fine overall concerning the language (although also the main text harbours some mistakes). The language mistakes distract from the content. I would recommend to review the article concerning language again carefully, with a special focus on the abstract.

Specific comments

Line 14 - it should be "toxins" (plural)

Line 14 and 16 - I would add a "the" before oral cavity

Line 17 and 18 - the sentecnce should be formulated in a way that is correct especially concerning the language (e.g., "it needs to be both somewhat insensitive as well as exquisitely sensitive ...." - or soemthing like that)

Line 20 and 21 - I would not delete the sentence in the brackets within the abstract. You describe later on which topics you will discuss and which not.

Table 1, 5. - orofacial pain (not pains)

Line 70 and 474 - I don`t know where Box 1 is. Two times "Box 1" is referenced, but I could not find a "Box 1" in the manuscript.

Line 78 - there is a space too much before "quantitative sensory testing"

Line 118 - projections should be written with a small letter.

Line 131 - in Wistar rats (plural).

Line 160/161 - "in rats" is written twice in this sentence - please omit one (preferably the second one).

Figure 2 - The unit of the x-axis is labelled as "s" (seconds I assume), but the legend says that the number of behavioral events was assessed. Please explain the discrepancy or change either the legend or the label on the axis.

Line 220 - if the paper is accepted for publication the last sentence does not make sense. I would delete it. If there is no reference, it would suggest that it is your own data anyway.

Line 228 - there is a space too much before "These".

Line 236 - is oral mucositis really the most debilitating adverse effect of cancer therapy? What about neuropathic pain, fatigue, nausea, etc.? I would suggest to soften the statement, e.g. oral mucositis is a debilitating adverse effect or is one of the most debilitating adverse effect or something like that.  

Line 271 - I think the abbreviation "ATP" has not been introduced yet. Abbreviations should be explained when first occuring. 

Line 294 - "hypesthesia" - do you mean hypoesthesia?

Line 299 - "...than patients are to TN..." this makes no sense. I assume you mean "PTTN patients are more resistant to therapy than patient with TN..." - please adjust.

Line 306 - "shows" (not show, since it corresponds to "topical administration")

Line 309 - Do you mean PTTN and TN? "Or" does not make sense in this case I think. Please adjust or explain.

Table 2 - Do you mean average age of disease onset?

Table 2 - "Rarely is trigger identifable" - Either "Rarely is a trigger identifiable" or "A trigger is rarely identified"

Line 322 - projects (not project - since it corresponds to "a subset")

Line 324 - Vc - The abbreviation is used earlier in the figure legend - so the explanation should be where Vc comes first (Line 321)

Line 331 - there is a space too much before "A rat"

Line 344 - "TG at 21 days" - please add "days" after 21.

Line 358-361 - This sentence needs revising. Maybe you mean "Non-nociceptive afferents express (more?) TRPV1 channels after injury...." - I think "become" does not belong within the sentence. But generally I am not entirely sure the sentence is well understandable. Please adjust.

Figure 4 - The red writing in A, B, F and G is not well readable. Maybe white would be better. 

Figure 4, E - the background color should probably be lighter (or more transparent) for better contrast. The lines and dashed lines are hard to see. Maybe a brighter background and bolder/wider lines would help.

Line 378 - A closing bracket is missing behind "ganglia".

Line 413-415 - For the quoted sentence a reference should be given.

Line 425/426 - The sentence needs revising. Do you mean "In BMS patients, TRPV1 expression is increased in the remaining nerve fibers and NGF expression is increased in nerve fibers and the basal epithelial layer"?

Line 431 - "These results suggest that peripheral..." - please add "that" to the sentence. 

Line 450 - It has been suggested that BMS should be (not just be) classified....Please revise.

Line 494 - Fos has not been introduced before. It should at least be explained shortly what Fos is and stands for.

Line 524 - "...need to be interpreted as neuroplasticity and could be..." - I think there is an "and" missing between neuroplasticity and could. Please revise.

Line 525 - somatosensory is written twice in a row. Please delete one.

Line 550 - macrophages (plural)

Author Response

Dear reviewer,

Thank you so much for reviewing our manuscript and spend your valuable time for that. And your comments were all very valuable and improve our manuscripts well from original. Thanks again for your great help and support!

Thanks,

Yu Shin

Reviewer 2 Report

Dear authors,

The study is a very interesting topic. Please check and revise written English.

For example:

Abstract section: "Therefore, it needs to be little bit insensitive but also exquisite sensitivity..." This sentence should be rewritten. "little bit" does not mean anything scientifically.

Author Response

Dear Reviewer,

Thank you so much for reviewing our manuscript to spend your valuable time. By your comments to improve our manuscript, our manuscript is much better than before. Thanks again for your positive review and great support!

Thanks,

Yu Shin

Reviewer 3 Report

I have no concern to publish this article in this journal.

Author Response

Dear Reviewer,

Thank you so much for reviewing our manuscript to spend your valuable time. Thanks again for your positive review and great support!

Thanks,

Yu Shin
